# Comparative Effect of Conventional and Non-Conventional Over-the-Counter Treatments for Male Androgenetic Alopecia: A Systematic Review and Network Meta-Analysis

**DOI:** 10.3390/ijms26167920

**Published:** 2025-08-16

**Authors:** Aditya K. Gupta, Mesbah Talukder, Sharon Keene, Greg Williams, Mary A. Bamimore

**Affiliations:** 1Division of Dermatology, Department of Medicine, Temerty Faculty of Medicine, University of Toronto, Toronto, ON M5S 3H2, Canada; 2Mediprobe Research Inc., London, ON N5X 2P1, Canada; mesbah.talukder@bracu.ac.bd (M.T.); mbamimore@mediproberesearch.com (M.A.B.); 3School of Pharmacy, BRAC University, Dhaka 1212, Bangladesh; 4Physician’s Hair Institute, Tucson, AZ 85712, USA; drkeene@hairrestore.com; 5Farjo Hair Institute, London W1G 7LH, UK; dr.greg@farjo.com

**Keywords:** male androgenetic alopecia, male-pattern hair loss, minoxidil, network meta-analysis, pathophysiology, over-the-counter products, conventional and non-conventional treatments

## Abstract

The expanding literature on nutraceuticals for male androgenetic alopecia (AGA) has inadvertently created knowledge gaps on the relative efficacy of conventional and non-conventional over-the-counter (OTC) treatments. We conducted a Bayesian network meta-analysis (NMA); the outcome measure was the 24-week change in total hair density (INPLASY202570087). We determined the relative efficacy of nine active comparators—including topical minoxidil 5% and 2%. Among the topical OTC agents used to manage male AGA, minoxidil 5% applied twice daily was the most effective.

## 1. Introduction

Androgenetic alopecia (AGA) is the most common form of hair loss in men [1]. It has both hereditary and androgen-dependent components. Male AGA is also referred to as male-pattern hair loss (MPHL). It was coined as a term in 1942 by Hamilton [2]. The Norwood–Hamilton scale—established in 1975—is the most commonly used classification system for male AGA [3].

Male AGA manifests as miniaturization of the hair follicle, the process whereby changes to the structural biology of the hair follicle cause it to stop producing terminal (i.e., non-vellus) hairs and transition to vellus hairs [1,4,5,6,7]. Hence, this change in the ‘terminal-to-vellus hair ratio’ is what results in the ‘phenotypic’ manifestation of patterned hair loss. While hair follicle miniaturization is an established manifestation of AGA, the etiology—or pathophysiology—behind this cause is yet to be fully understood; notwithstanding this, androgens are still a major culprit behind the miniaturization. The binding of dihydrotestosterone (DHT) to androgen receptors of the hair follicle triggers miniaturization. This is why long-standing conventional therapeutic agents for AGA—such as finasteride—were formulated to target androgen biochemistry. Finasteride inhibits the Type II isoenzyme of 5 alpha-reductase that catalyzes the conversion of testosterone to DHT, which binds to the androgen receptor (AR). The DHT-AR complex acts as a transcription factor in the cell nucleus, where it triggers a sequence of events resulting in the inhibition of the Wnt/β-catenin signaling and subsequent miniaturization of hair, manifesting as hair loss.

The pathophysiology of male AGA may not be entirely androgen-dependent because it has a multifactorial etiology, and a variety of agents putatively not acting through the testosterone-DHT pathway have shown efficacy in treating AGA. For example, minoxidil, a vasodilator, has demonstrated efficacy in male AGA [8,9].

The response to topical minoxidil varies, partly due to genetic factors such as sulfotransferase 1A1 (SULT1A1) enzyme activity, which converts minoxidil (pro-drug) to its active form, minoxidil sulfate [10,11]. Low SULT1A1 activity is associated with non-response or poor response, affecting a significant subset of patients and potentially confounding clinical trial outcomes [12]. Minoxidil has also been theorized to target androgen biochemistry in AGA [13,14]. For instance, this agent may disrupt transcription of the androgen receptor gene [13].

The peer-reviewed male AGA literature constitutes published empirical studies on the therapeutic impact of conventional versus non-conventional over-the-counter (OTC) agents. The current study has determined the comparative effectiveness of conventional and non-conventional OTC products for male AGA through a network meta-analysis (NMA). To our knowledge, such an NMA has not been previously reported.

## 2. Materials and Methods

On March 30, 2025, we systematically reviewed the peer-reviewed literature through PubMed and Scopus to identify relevant data for our NMA, and the entire conduct thereof was in accordance with the Preferred Reporting Items for Systematic reviews and Meta-Analyses (PRISMA) guidelines for NMAs (protocol registration: INPLASY202570087) [15].

Studies that were eligible for our network had to be published in English and be of a trial design with an arm investigating the effect of monotherapy with relevant conventional or non-conventional OTC treatments for male AGA. As per the PICO (i.e., population, intervention, comparator, outcome) framework for eligible studies, the population of interest was men with AGA; the intervention and/or comparator was the relevant conventional or non-conventional OTC therapeutic agent for male AGA; and the outcome of interest was change in total hair density 24 weeks from baseline (measured in hairs per cm^2^).

The detailed search strategy indicated above was guided by the recently published peer-reviewed literature [16,17,18,19,20] on alternative and complementary therapies for androgenetic alopecia. The complete search strategy, including Medical Subject Headings (MeSH) terms, is provided in Appendix A.

We excluded various studies that did not meet the sex criterion of our eligibility criteria; our NMA was specific for the male sub-population of persons with androgenetic alopecia (AGA). The efficacy of numerous over-the-counter (OTC) interventions not approved by the Food and Drug Administration (FDA) for AGA did not report data separately for men, and our network meta-analysis (NMA) was specific for the male population. For instance, the studies by Piquero-Casals et al. (2025) [21] and De Biasio et al. (2023) [22] were—despite being of a randomized double-blinded design—excluded from our NMA because the research participants of the respective studies were of the male and female sexes. Furthermore, outcome data in these studies were not reported separately for males and females. Piquero-Casals et al. (2025) [21] compared the efficacy of saw palmetto (oral) with placebo, while De Biasio et al. (2023) [22] investigated the impact of procyanidin (oral) relative to placebo. The sex criterion was the same reason we also randomized double-blinded trials that investigated the FDA-approved OTC treatments, including those by Bergfeld et al. (2016) [23] (who investigated the efficacy of minoxidil 5% (topical)).

Trials were also excluded because they did not include our outcome measure of interest insofar as the timepoint; our NMA included trials that quantified our outcome of interest (i.e., change in total hair density) at 24 weeks (or 6 months). Trials that investigated combination therapy were ineligible for our NMA.

At the time of our review, expert opinion suggested that our NMA include ketoconazole, melatonin, saw palmetto, and rosemary as comparators. Therefore, we mined references to identify trials that investigated these agents while minimally relaxing our criteria. Hence, we found 4 trials that still investigate men with AGA and used the 24-week change in total hair density as an outcome measure.

The stages of screening titles, abstracts, and full texts were performed independently by two authors (MAB and MT). Any disagreements were resolved through discussion with a third author (AKG). The screening stages were managed with the Rayyan web application [24]. Extracted data were organized in spreadsheets.

The evidence quality of each eligible study was qualitatively assessed using the Cochrane Collaboration’s Risk of Bias tools, where qualitative evaluations were summarized with traffic plots [25].

We constructed a network plot to depict direct evidence; a network plot is a graph of nodes and edges where a node corresponds to an intervention (or comparator) and is represented as a vertex [26]. An edge represents a line between two nodes and corresponds to the direct comparison of the two comparators’ effect in an actual head-to-head trial. The geometry of the plot also determined whether a node-splitting analysis for inconsistency could ensue. The consistency between direct and indirect evidence can be statistically assessed with the node-splitting analysis, where the null hypothesis (i.e., the hypothesis that states there is no inconsistency) is rejected when *p*-values are significant (i.e., *p* < 0.05). Evidence of no inconsistency supports that the network is transitive [27,28].

All analyses were conducted using RStudio version 2023.9.1.494 (R version 4.3.2) [28]; the multinma package (version 0.6.0) [29] was used. We ran a Bayesian NMA under a fixed effects model with uniform priors. The NMA produced surface under the cumulative ranking curve (SUCRA) values for each comparator, and an intervention’s SUCRA value ranges between 0 to 1 (or 0% to 100%). An NMA also produces estimates of pairwise relative effects across a network. Given the outcome of interest, the point estimate for the pairwise effect is the mean difference (MD). We reported MDs along with their 95% credible intervals (CIs). In addition to our NMA (i.e., base analysis), we conducted a sensitivity analysis adjusted for variation in disease severity.

## 3. Results

Through our systematic search, we found 15 relevant trials that met our eligibility criteria, across which nine active comparators were identified, namely, (1) ketoconazole 2% (topical), (2) marine complex (oral), (3) minoxidil 2% (topical), (4) minoxidil 5% (topical), (5) procyanidin 0.7% (topical), (6) rosemary (topical), (7) saw palmetto (topical), (8) melatonin (topical), and (9) watercress 2% (topical). Table 1 indicates the frequency of the active comparators or interventions. The node for ‘Control’ (i.e., the 10th comparator) amalgamated the vehicle and placebo arms across the included studies. Arm-level study characteristics of the 15 trials are provided in Table 1. The search process for the identification of eligible studies is summarized in the flow chart in Figure 1. The summary for our qualitative evaluation of evidence quality is outlined in Figure 2. The network plot is presented in Figure 3; the geometry of the plot did not permit the conduct of a node-splitting analysis for inconsistency. The kilim plot in Figure 4 presents comparators’ SUCRA values from the base and the two sensitivity analyses. Pairwise relative effects—from the base analysis—are summarized in the league table in Figure 5.

The eligible studies’ year of publication ranged from 1986 to 2022; 12 of the 15 eligible studies were randomized, of which 11 were double-blinded (Table 1). We deemed the network’s evidence quality to be moderate as most of the constituent studies were randomized and double-blinded (Table 1, Figure 2). Our network plot is depicted in Figure 3, and the geometry thereof precluded the conduct of a node-splitting analysis [45].

Our analysis showed that minoxidil 5% (topical) 1 mL twice daily was the most effective option (SUCRA = 99.8%), followed by melatonin (topical) (SUCRA = 61.9%). Minoxidil 5% (topical) was more effective than melatonin (topical) (MD = 18.96 hairs/cm^2^, 95% CI: (3.51, 35.29) hairs/cm^2^, *p* < 0.05) and rosemary (topical) (MD = 19.53 hairs/cm^2^, 95% CI: (6.52, 32.68) hairs/cm^2^, *p* < 0.05). In our network, minoxidil 5% (topical) 1 mL twice daily was more effective than minoxidil 2% 1 mL twice daily (topical) (MD = 20.48 hairs/cm^2^, 95% CI: (10.94, 30.21) hairs/cm^2^, *p* < 0.05). Minoxidil 5% (topical) was also more effective than saw palmetto (topical) (MD = 30.91 hairs/cm^2^, 95% CI: (13.73, 48.76) hairs/cm^2^, *p* < 0.05). Rosemary (topical) was more efficacious than control (MD = 10.63 hairs/cm^2^, 95% CI: (0.82, 20.89) hairs/cm^2^, *p* < 0.05). Similarly, marine complex (topical) was more effective than control (MD = 9.73 hairs/cm^2^, 95% CI: (3.81, 15.67) hairs/cm^2^, *p* < 0.05) (Figure 4 and Figure 5).

## 4. Discussion

Investigating the therapeutic potential of non-conventional agents to treat male pattern AGA is a worthwhile endeavor because many consumers—rightly or wrongly—believe in non-conventional medicine for both cosmetic and non-cosmetic conditions [46,47]. Moreover, the salience of such investigation is increased by the concerns raised by the rise in adverse event reporting with conventional treatments for AGA, such as oral finasteride. Another benefit of conducting more studies on non-conventional agents would be the increased opportunity to collect more safety data. Table 2 represents potential mechanisms of action of OTC products in promoting hair growth. Our analysis showed that the FDA-approved minoxidil 5% (topical) was significantly more effective, as measured by the 24-week change in total hair count, than melatonin (topical), rosemary (topical), marine complex, minoxidil 2% (topical), procyanidin 0.7% (topical), watercress 2% (topical), ketoconazole 2% (topical), and saw palmetto (topical).

The findings are consistent with the pathogenesis of male AGA having both androgen-dependent (e.g., saw palmetto’s 5-α-reductase inhibition) and androgen-independent (e.g., minoxidil’s vasodilation) pathways. Our study confirms that topical minoxidil 5% 1 mL twice daily is the most efficacious FDA-approved topical OTC for male AGA, while melatonin shows promising potential, warranting further investigation for AGA management.

### 4.1. SULT1A1 and Minoxidil Response

The absence of SULT1A1 activity assessment in the included trials is an important limitation. SULT1A1 converts minoxidil (a pro-drug) to its active sulfate form. Low activity of SULT1A1, prevalent in 20–30% of male AGA patients, is linked to non-response [10,12]. A 2019 study validated a SULT1A1 enzymatic assay predicting minoxidil response with 95% sensitivity in men with AGA [48]. Without stratifying patients by SULT1A1 activity, the efficacy of minoxidil in our NMA likely underestimates population-level effects, as non-responders dilute the responder subgroup’s outcomes. This impacts statistical point estimation, potentially confounding minoxidil’s true efficacy in SULT1A1-high patients by 20–40% [49].

Future trials should incorporate SULT1A1 testing (via scalp biopsy or genetic screening) to refine efficacy estimates and enable personalized treatment. This omission also affects comparisons with other non-conventional interventions, which may appear relatively more effective due to unadjusted minoxidil data.

### 4.2. Melatonin as an AGA Treatment

Melatonin, a neurohormone with antioxidant, anti-inflammatory, and hair growth-promoting properties, shows promise for AGA [38,50]. These effects are mediated by melatonin’s activation of Wnt/β-catenin signaling. Melatonin’s moderate efficacy and favorable safety profile (minimal side effects, e.g., mild scalp irritation for topical, drowsiness for oral) make it a viable adjunctive treatment, particularly for patients who are intolerant to minoxidil or seeking non-androgen-targeted therapies. However, its lower efficacy compared to minoxidil 5% (topical) limits its use as a primary treatment. More RCTs are needed to optimize dosing (e.g., higher topical concentrations) and compare melatonin directly with topical minoxidil in SULT1A1-stratified populations.

### 4.3. Limitations

The absence of SULT1A1 data in minoxidil trials reduces the accuracy of efficacy estimates, potentially by 20–40% in non-responders [49]. For many of the non-conventional OTC products, there are a limited number of trials with small numbers of subjects in the trials. Additionally, the effectiveness of topical treatments may be influenced by several factors, such as the specific application method, scalp condition (e.g., dryness or oiliness), and whether the scalp is wet or dry during application. These confounding factors may impact the findings of our NMA study. Additionally, several high-quality studies were excluded because they did not meet the specific methodological criteria used in this study.

## 5. Conclusion

Minoxidil 5% (topical), an FDA-approved therapy for male AGA, remains an effective topical treatment. Other non-conventional OTC products are topical preparations of melatonin and rosemary oil. Future research should consider RCTs treating larger numbers of male AGA patients to gain a better understanding of the efficacy and safety of non-conventional OTC agents.

Our NMA is consistent with the etiology of male androgenetic alopecia involving both androgen-dependent and androgen-independent mechanisms. SULT1A1 testing could help personalize minoxidil therapy, improving efficacy estimates and treatment outcomes. Further research is warranted to substantiate these observations.

## Figures and Tables

**Figure 1 ijms-26-07920-f001:**
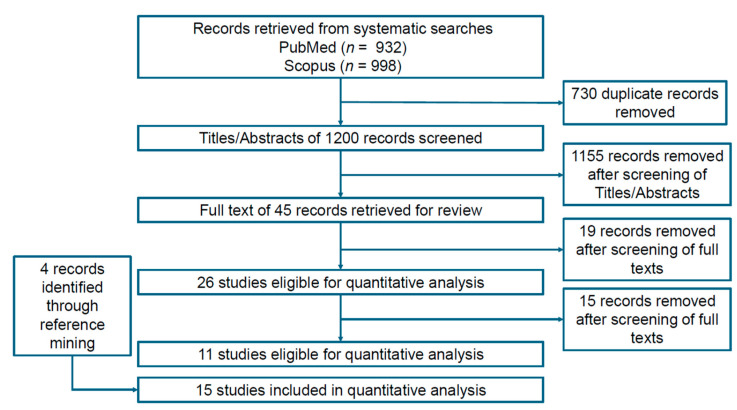
Schematic for identification of included studies. Herein, the search process for identification of eligible studies is summarized in this flow chart.

**Figure 2 ijms-26-07920-f002:**
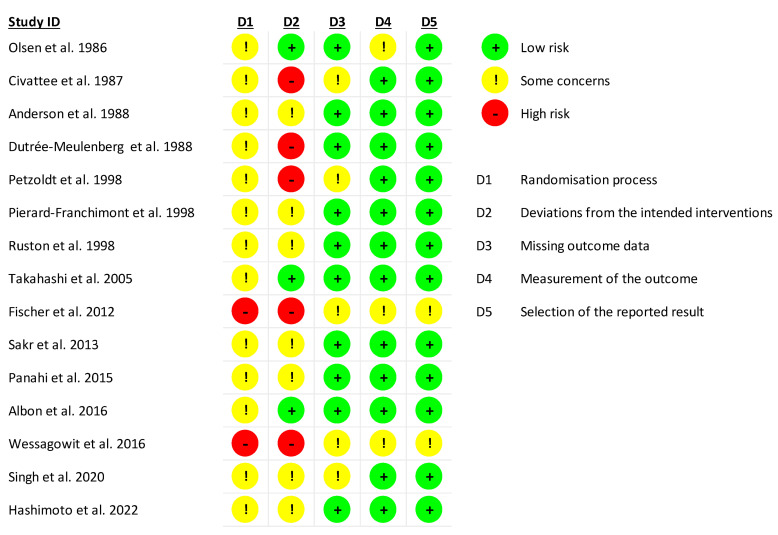
Arm-level evaluation of evidence quality. In this traffic plot, the colors green, yellow, and red indicate low, moderate, and high risk of bias, respectively. D1 to D5 correspond to the various domains for risk of bias: D1 refers to bias arising from the randomization process, D2 refers to bias due to deviations from intended intervention, D3 refers to bias due to missing outcome data, D4 refers to bias in measurement of the outcome, and D5 refers to bias in selection of the reported result [30,31,32,33,34,35,36,37,38,39,40,41,42,43,44].

**Figure 3 ijms-26-07920-f003:**
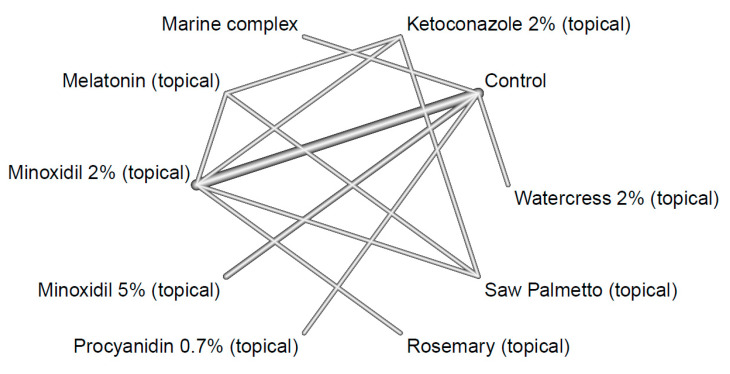
Network plot. This graph represents the connection of comparators across our network for 24-week change in total hair density. The thickness of the line indicates the number of studies. The thicker the line, the greater the number of studies.

**Figure 4 ijms-26-07920-f004:**
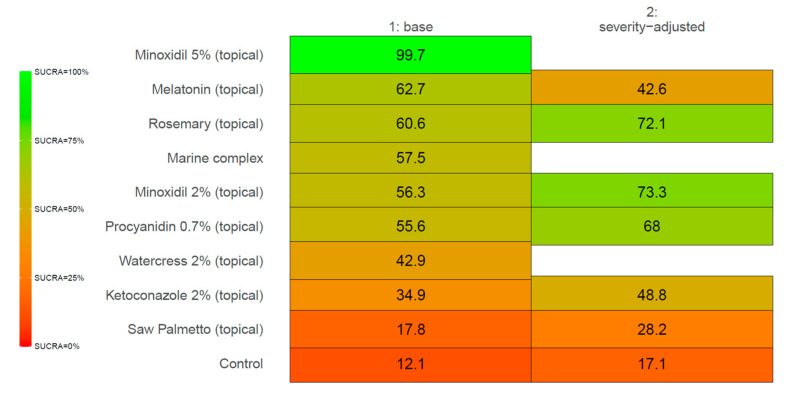
Kilim plot for efficacy rankings. The Surface Under the Cumulative Ranking Curve (SUCRA) values of a comparator indicate its overall efficacy ranking, with higher values signifying a more favorable impact. Vertical Axis: Represents the comparators; our network included nine active comparators and a control node that combined placebo and vehicle treatments. Horizontal Axis: Displays endpoints, with the first and second columns illustrating SUCRA values from the base and severity-adjusted, respectively. The SUCRA values in these columns are expressed as percentages. Color Chart: A visual representation of SUCRA values is included; colors green and red (chosen arbitrarily) denote the highest and lowest SUCRA values, respectively. This illustrates the treatments ranked from most to least effective based on this metric. The blank cells indicate that the data were not available.

**Figure 5 ijms-26-07920-f005:**
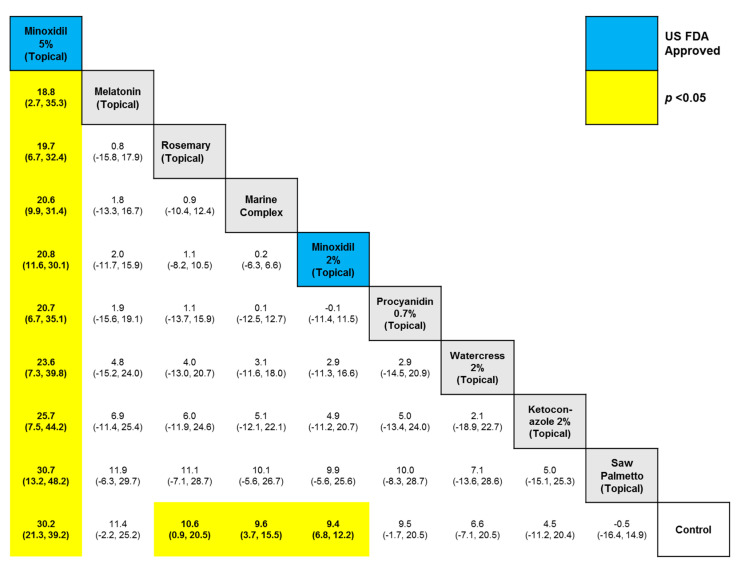
League table. Relative effects from every possible pairwise comparison are presented herein as the mean difference (MD); in parentheses are the corresponding 95% credible interval (CI).

**Table 1 ijms-26-07920-t001:** Arm-level summary of characteristics of included studies.

Author and Year	Study Design	Treatment	Frequency of Use	Sample size at Baseline	Age, Standard Deviation ^ͳ^)	Disease Severity (Mild, Moderate, Severe) ^†^
Olsen 1986 [30]	randomized, double-blind	Minoxidil 2% (topical)	1 mL twice daily	19	35.4, 4	0%, 100%, 0%
		Control		19	37, 6.3	0%, 100%, 0%
Civatte 1987 [31]	randomized, double-blind	Minoxidil 2% (topical)	1 mL twice daily	125	33.4 (18–49)	0%, 48.15%, 51.85%
		Control		122	34.8 (20–40)	0%, 55.29%, 44.71%
Anderson 1988 [32]	randomized, double-blind	Minoxidil 2% (topical)	1 mL twice daily	77	32.8 (19–49)	0%, 100%, 0%
		Control		77	32.8 (19–49)	0%, 100%, 0%
Dutrée-Meulenberg 1988 [33]	randomized, double-blind	Minoxidil 2% (topical)	1 mL twice daily	74	34.3 (19–49)	
		Control		70	34.3 (19–49)	
Petzoldt 1988 [34]	randomized, double-blind	Minoxidil 2% (topical)	1 mL twice daily	101	32.9 (17–49)	
		Control		100	32.9 (17–49)	
Pierard-Franchimont 1998 [35]	prospective cohort	Minoxidil 2% (topical)	2 to 4 times weekly for 21 months	4		0%, 100%, 0%
		Ketoconazole 2% (topical)		4		0%, 100%, 0%
Rushton 1998 [36]	randomized, double-blind	Minoxidil 2% (topical)	1 mL twice daily	12	18–49	
		Control		12	18–49	
Takahashi 2005 [37]	randomized, double-blind	Procyanidin 0.7% (topical)	2 mL twice daily	25		47.62%, 52.38%, 0%
		Control		24		59.09%, 40.91%, 0%
Fischer 2012 [38]	single-arm	Melatonin (topical)	once daily	35	18–41	100%, 0%, 0%
Sakr 2013 [39]	randomized, double-blind	Minoxidil 5% (topical)	1 mL twice daily	11	25–30	
		Control		10	25–30	
Panahi 2015 [40]	randomized, single-blind	Rosemary (topical)	1 mL twice daily	50	24.78, 3.67	74%, 26%, 0%
		Minoxidil 2% (topical)		50	23.38, 2.5	68%, 32%, 0%
Ablon 2016 [41]	randomized, double-blind	Marine complex (oral)	one capsule per day	30	42.8, 7.7	
		Control		30	46.1	
Wessagowit 2016 [42]	single-arm	Saw Palmetto (topical)	once daily	100	20–50	0%, 70%, 30%
Singh 2020 [43]	randomized, double-blind	Minoxidil 5% (topical)	1 mL twice daily	20	18–60	
		Control		20	18–60	
Hashimoto 2022 [44]	randomized, double-blind	Watercress 2% (topical)	twice daily	23		
		Control		23		

Notes: ^ͳ^ We reported the range if the authors did not provide the mean age or standard deviation. ^†^ Disease severity was categorized as per the Norwood–Hamilton scale. With expert opinion, we deemed mild to correspond to Stages I and II, moderate to correspond to Stages III and IV, and severe to correspond to Stages V and above.

**Table 2 ijms-26-07920-t002:** Potential mechanisms of action of over-the-counter (OTC) agents in promoting hair growth.

Agent	Mechanism of action
Minoxidil	Vasodilator Reduce inflammation Activate Wnt/β-catenin signaling pathway
Melatonin	Activate Wnt/β-catenin signaling pathway Antioxidant Anti-inflammatory properties Anti-androgen properties
Rosemary	Antioxidant Antibacterial Antifungal Anti-inflammatory properties
Marine complex	Stimulate proliferation of dermal papilla cells
Procyanidin	Reduction of oxidative stress and inflammation Encourage growth of hair epithelial cells
Watercress	Modulate key signaling pathways (notably Wnt/β-catenin via RSPO1 and DKK1) Activating growth-promoting kinases Antioxidant property
Ketoconazole	Anti-inflammatory Anti-androgen properties
Saw Palmetto	Inhibition of 5-α-reductase enzyme

Wnt/β-catenin: Wingless/integrated beta-catenin signaling pathway; RSPO1: R-spondin 1; DKK1: Dickkopf-related protein 1.

## Data Availability

The raw data supporting the conclusions of this article will be made available by the authors on request.

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
