# Peer review of "Comparative Effect of Conventional and Non-Conventional Over-the-Counter Treatments for Male Androgenetic Alopecia: A Network Meta-Analysis Study"

_ijms, 2025, doi:10.3390/ijms26167920_

Round 1
Reviewer 1 Report
Comments and Suggestions for Authors
- Please clarify the search that was conducted (also as supplementary material) specifying the mesh terms used. It is not clear why only these few components were evaluated considering how many products over-the-counter exist.
- the search should be re-run considering it was done more than one year ago
- the title says: "Etiologic basis of male androgenetic alopecia.." but no one of the study report soldi and comparable data regarding etiologic basis of AGA. I think the title should be limited to the efficacy reported
Author Response
Reviewer-1
Comment: Please clarify the search that was conducted (also as supplementary material) specifying the mesh terms used. It is not clear why only these few components were evaluated considering how many products over-the-counter exist.
Response: We appreciate the reviewer’s comment. In response, the detailed search strategy has been included in the manuscript.
“Full search strategy with Medical Subject Heading (MeSH) terms included:
(((((((trial*) OR (effec*)) OR (effic*)) OR (impact*)) OR (random*)) OR (stud*)) OR (ex-per*) AND (1000/1/1:2025/3/30[pdat]) AND (1000/1/1:2025/3/30[pdat])) AND ((((((((((Procyanidin*[Title/Abstract]) OR (Cetirizine*[Title/Abstract])) OR (Caf-feine*[Title/Abstract])) OR ("pumpkin seed"[Title/Abstract])) OR ("bran"[Title/Abstract]) ) OR ("rosemary"[Title/Abstract])) OR ("saw palmet-to"[Title/Abstract])) OR ("watercress"[Title/Abstract])) OR (Alternative Thera-pies[MeSH Terms])) AND ((hair loss) OR ((Hair loss[MeSH Terms]) OR (alopecia* OR baldness*))))
Studies that were eligible for our network had to: be published in English and be of a trial design with an arm investigating the effect of monotherapy with relevant conventional or non-conventional OTC treatments for male AGA. As per the PICO (i.e., population, intervention, comparator, outcome) framework for eligible studies, the population of interest was men with AGA; the intervention and/or comparator was the relevant conventional or non-conventional OTC therapeutic agent for male AGA; and the outcome of interest was change in total hair density 24 weeks from baseline (measured in hairs per cm2).
The detailed search strategy indicated above was guided by the recently published peer-reviewed literature[16-20] on alternative and complementary therapies for androgenetic alopecia.
Reasons for exclusion (a detailed summary)
Trials were excluded from our network meta-analysis (NMA) if they did not meet any aspects of our eligibility criteria.
We excluded various studies that did not meet the sex criterion of our eligibility criteria; our NMA was specific for the male sub-population of persons with androge-netic alopecia (AGA). The efficacy of numerous over-the-counter (OTC) interventions not approved by the Food and Drug Administration (FDA) for AGA did not report data separately for men and our network meta-analysis (NMA) was specific for the male population. For instance, the studies by Piquero-Casals et al. (2025)[21] and De Biasio et al. (2023)[22] were—despite being of a randomized double-blinded design—excluded from our NMA because the research participants of the respective studies were of the male and female sexes. Furthermore, outcome data in these studies were not reported separately for males and females. Piquero-Casals et al. (2025)[21] compared the effica-cy of saw palmetto (oral) with placebo, while De Biasio et al. (2023)[22] investigated the impact of procyanidin (oral) relative to placebo. The sex criterion was the same reason we also randomized double-blinded trials that investigated the FDA-approved OTC treatments, including those by Bergfeld et al. (2016)[23] (who investigated the ef-ficacy of minoxidil 5% (topical).
Trials were also excluded because they did not include our outcome measure of interest insofar as the timepoint; our NMA included trials that quantified our outcome of interest (i.e., change in total hair density) at 24 weeks (or 6 months). Hence, we ex-cluded the randomized trials by Ibrahim et al. (2021) [24], (who investigated the efficacy of pumpkin seed oil (topical) relative to minoxidil 5% (topical) and Choi et al. (2015) [25], (who studies the impact of rice bran (topical) relative to placebo), to name a few.
Trials that investigated combination therapy were ineligible for our NMA. Hence, we excluded the randomized trials by Davis et al. (2021) [26] (who investigated the impact of caffeine combined with niacinamide and zinc pyrithione, among other things) and Welzel et al. (2022) [27], (who investigated the efficacy of polytherapy with caffeine), to name a few.
At the time of our review, expert opinion suggested that our NMA include ketoconazole, melatonin, saw palmetto and rosemary as comparators. So, we mined references to identify trials that investigated these agents while minimally relaxing our criteria. Hence, we found 4 trials that still investigate men with AGA and used 24-week change in total hair density as an outcome measure.”
Page: 2-3; Lines: 69-122
Comment: The search should be re-run considering it was done more than one year ago
Response: We thank the reviewer for pointing that out. It was a typographical error — the correct date is March 30, 2025. The manuscript has been updated accordingly.
Page:2; Line: 63
Comment: the title says: Etiologic basis of male androgenetic alopecia." but no one of the study report soldi and comparable data regarding etiologic basis of AGA. I think the title should be limited to the efficacy reported
Response: We appreciate the reviewer’s comment. In response, the title has been amended to: “Comparative effect of conventional and non-conventional over-the-counter treatments for male androgenetic alopecia: A synthesis of the literature and a network meta-analysis study.”
Page: 2; Line: 63
Reviewer 2 Report
Comments and Suggestions for Authors
It appears there are some significant studies that are missing from this network meta-analysis, for example the pivotal trials for 5% minoxidil: Olsen EA, Dunlap FE, Funicella T, Koperski JA, Swinehart JM, Tschen EH, Trancik RJ. A randomized clinical trial of 5% topical minoxidil versus 2% topical minoxidil and placebo in the treatment of androgenetic alopecia in men. J Am Acad Dermatol. 2002 Sep;47(3):377-85. doi: 10.1067/mjd.2002.124088. PMID: 12196747.
The number of patients reviewed overall seems low and the fact that topical melatonin scored so highly is suspect as a lower quality study.
Overall, I do like the spirit of this study to compare over the counter treatment options for male AGA, but I am not confident that the results are accurate.
Also, I am uncertain that this study can confidently say that there are androgen-dependent and androgen-independent etiologies for male AGA simply based on the response to these treatments. While the mechanism of action of minoxidil is thought to be related to its vasodilatory effects, it is not completely clear that is how it works. While few, there are some studies suggesting an anti-androgen effect from minoxidil.
Author Response
Reviewer-2
Comment: It appears there are some significant studies that are missing from this network meta-analysis, for example the pivotal trials for 5% minoxidil: Olsen EA, Dunlap FE, Funicella T, Koperski JA, Swinehart JM, Tschen EH, Trancik RJ. A randomized clinical trial of 5% topical minoxidil versus 2% topical minoxidil and placebo in the treatment of androgenetic alopecia in men. J Am Acad Dermatol. 2002 Sep;47(3):377-85. doi: 10.1067/mjd.2002.124088. PMID: 12196747.
Response: We thank the reviewer for the valuable comment. However, the included studies were selected based on the predefined inclusion and exclusion criteria outlined in the Methods section of the manuscript. For example, the study by Olsen et al. (2002) was excluded because it assessed treatment efficacy at 48 weeks, whereas our network meta-analysis focused on changes in total hair density at 24 weeks from baseline. This time point was chosen to ensure consistency across studies evaluating both conventional and non-conventional over-the-counter treatments.
Comment: The number of patients reviewed overall seems low and the fact that topical melatonin scored so highly is suspect as a lower quality study.
Response: We appreciate the reviewer’s comment. We agree with the reviewer that the strength of evidence for melatonin may be limited by the lower quality and small sample sizes (n = 35) of the available studies. In response, the following sentence has been added to the manuscript. However, we would like to add that a recent comprehensive review by Tawanwongsri and Eden (2025), which supports the potential efficacy and safety profile of topical melatonin.
“Future research should consider RCTs treating larger numbers of male AGA patients to gain a better understanding the efficacy and safety of non-conventional OTC agents.”
Page:12; Lines: 272-273
Comment: Overall, I do like the spirit of this study to compare over the counter treatment options for male AGA, but I am not confident that the results are accurate.
Response: We understand the reviewer’s concerns and recognize the inherent limitations in the available evidence base, including variability in study design and sample sizes. We have taken careful measures to apply rigorous inclusion criteria and transparent methodology, to enhance the reliability of our findings. Nonetheless, we agree that continued high-quality research in this area is essential.
“Future research should consider RCTs treating larger numbers of male AGA patients to gain a better understanding the efficacy and safety of non-conventional OTC agents.”
Page:12; Lines: 272-273
Comment: Also, I am uncertain that this study can confidently say that there are androgen-dependent and androgen-independent etiologies for male AGA simply based on the response to these treatments. While the mechanism of action of minoxidil is thought to be related to its vasodilatory effects, it is not completely clear that is how it works. While few, there are some studies suggesting an anti-androgen effect from minoxidil.
Response: We thank the reviewer for the comment. In response, the manuscript (abstract and conclusion) has been updated accordingly.
“Our NMA is consistent with the etiology of male androgenetic alopecia involving both androgen-dependent and androgen-independent mechanisms. SULT1A1 testing could help personalize minoxidil therapy, improving efficacy estimates and treatment outcomes. Further research is warranted to substantiate these observations.”
Page: 12; Lines: 275-278
Round 2
Reviewer 2 Report
Comments and Suggestions for Authors
I appreciate the detailed explanation regarding exclusion of many studies. I believe it can be shortened, however. It can also be added to the discussion/limitations section as a reminder that there are many studies of good quality that were not included because of the methodology of this study.
Author Response
Reviewer-1
Comment: I appreciate the detailed explanation regarding exclusion of many studies. I believe it can be shortened, however. It can also be added to the discussion/limitations section as a reminder that there are many studies of good quality that were not included because of the methodology of this study.
Response: We appreciate the reviewer’s comment. In response:
- We have placed the “Full search strategy with Medical Subject Heading (MeSH) terms” section at the end of the paper, after the Conclusion, to improve readability of the manuscript.
- We have substantially shortened the Materials and Methods section, reducing it from 937 words to 777 words.
- We have added the following sentence to the Limitations section: “Additionally, several high-quality studies were excluded because they did not meet the specific methodological criteria used in this study.”
Page: 2-3, 11; Lines: 63-101, 248-249, 260-267